# Comparison of Effects of N95 Respirators and Surgical Masks to Physiological and Psychological Health among Healthcare Workers: A Randomized Controlled Trial

**DOI:** 10.3390/ijerph182413308

**Published:** 2021-12-17

**Authors:** Che-Yu Su, Chiung-Yu Peng, Hsin-Liang Liu, I-Jeng Yeh, Chi-Wei Lee

**Affiliations:** 1Department of Public Health, College of Health Science, Kaohsiung Medical University, Kaohsiung 80708, Taiwan; money0967@gmail.com (C.-Y.S.); pengcy@kmu.edu.tw (C.-Y.P.); 2Department of Emergency Medicine, Kaohsiung Medical University Hospital, Kaohsiung Medical University, Kaohsiung 80708, Taiwan; cutelpliu@gmail.com (H.-L.L.); ijengyeh1@gmail.com (I.-J.Y.); 3School of Medicine, College of Medicine, Kaohsiung Medical University, Kaohsiung 80708, Taiwan; 4Institute of Medical Science and Technology, National Sun Yat-sen University, Kaohsiung 80424, Taiwan

**Keywords:** N95 respirator, physiological parameters, psychological health, healthcare workers, Tensor Tip MTX device

## Abstract

Since the onset of the coronavirus disease 2019 pandemic, wearing facemasks has become more important for healthcare workers. This study aimed to investigate and compare the influence of wearing N95 respirators and surgical masks for 8 h on physiological and psychological health. Sixty-eight healthcare workers were randomly assigned to the N95 respirator or surgical mask groups. Physiological parameters of participants were measured by Tensor Tip MTX at baseline and at the 2nd, 4th, 6th and 8th h of wearing the facemasks. The symptoms after wearing facemasks were also determined via the questionnaire. There were no significant changes in physiological parameters at most time checkpoints in both groups. Significant differences were observed in terms of heart rate at the 8th h, time trends (adjusted difference of least squares means were −8.53 and −2.01), and interaction of time and mask type between the two groups (*p*-value for interaction was 0.0146). The values of these physiological parameters were within normal ranges. The N95 respirator group had significantly higher incidences of shortness of breath, headache, dizziness, difficulty talking and fatigue that spontaneously resolved. In conclusion, healthcare workers who wore either N95 respirators or surgical masks during an 8 h shift had no obvious harmful effects on physiological and psychological health. Additionally, the N95 respirator group did not show a higher risk than the surgical mask group.

## 1. Introduction

Since the onset of the coronavirus disease 2019 (COVID-19) pandemic, as declared by the World Health Organization (WHO) on 11 March 2020, over 225,000,000 people have been infected worldwide, with more than 4,640,000 deaths as of 15 September 2021 [1]. Many patients with symptoms associated with acute upper airway infection, such as fever, headache, cough, coryza, or sore throat, were highly suspected to be COVID-19 victims after visiting the hospitals every day and had caused much stress to the healthcare workers who contacted them. As the pandemic persists, wearing personal protective equipment (PPE) has become more important to healthcare workers for self-protection due to long-term and frequent contact with high-risk patients, especially in the emergency department (ED). In accordance with a report from a tertiary center in Wuhan, China in January 2020, twenty-nine percent (40/138) of patients who were admitted to the hospital for COVID-19 infection were healthcare workers, and most of them were victims of nosocomial infections that occurred after contacting with patients [2]. According to previous experience in dealing with influenza and severe acute respiratory syndrome (SARS), healthcare workers confronting different risks are recommended by the WHO to use different PPE; those who are responsible for non-aerosol-generating care should wear surgical masks and those who work on aerosol-generating or high-risk procedures, such as airway fluid suctioning or endotracheal intubation for patients with respiratory failure, should wear N95 respirators [3]. 

By definition, aerosols are suspensions of extremely small particles or droplets in the air with diameters of 0.01–100 µm. People to whom an aerosol stimulus was exposed may suffer from damage to their health and even contract infectious disease [4]. Most scholars believe that COVID-19 viruses were transmitted mainly through large droplets from the airway via close interpersonal contact [5]. The N95 respirator is designed to fit tightly into the human facial contour, providing prevention against inhalation of small airborne particles. In contrast, the surgical mask is made for loose fit to the human facial contour and protects mainly against large droplets especially exhaled aerosols from the wearers’ airway. [6]. According to current WHO recommendations, surgical masks are sufficiently safe for healthcare workers who are responsible for non-aerosol-generating care against COVID-19. [7,8]. However, the Centers for Disease Control and Prevention (CDC) of the United States of America and European Centre for Disease Control and Prevention (ECDC) both suggest donning the N95 respirator for daily routine duty, regardless of risks from clinical works of different nature [9,10].

According to the reported experience of healthcare workers working in the hospital, long-term wearing of facemasks caused symptoms such as headache, lightheadedness, lack of concentration, and anxiety brought about a concern about whether there are any adverse effects of long-term wearing of facemasks on healthcare workers [11,12,13]. Hypoxia or hypercapnia has manifested after long-term wearing of facemasks, especially N95 respirators, which, in turn, may affect the care quality of patients and health of healthcare workers [14,15]. Unfortunately, few studies have investigated the dynamic changes in physiological parameters among healthcare workers who wear facemasks for daily duties because of the infeasibility of obtaining arterial blood gas samples for scientific analysis because of the high risk of physical harm, such as bleeding or ecchymosis of soft tissue or skin from invasive procedures and disturbance of their daily duties. Whether wearing of either N95 respirators or surgical masks among healthcare workers for an 8 h daily duty had harmful effects on physiological and psychological health was always a controversial issue, and the objective of this study is to provide clarifying evidences through a scientific approach.

## 2. Materials and Methods

### 2.1. Study Design 

Healthcare workers, including nurses and doctors at the ED of the Kaohsiung Medical University Hospital (KMUH), a tertiary center in southern Taiwan, were recruited. Their duties rotated around the triage station, visiting zone, or critical zone. According to the guidelines, healthcare workers working at the triage station outside the hospital are recommended to wear N95 respirators (defined as N95 filtering face piece respirators certified by the National Institute for Occupational Safety and Health (NIOSH) and European standard) because of the high risk of infection from unclear travel, occupational, contact, and cluster patient histories [16]. In contrast, healthcare workers who work at the triage station inside the hospital are advised to wear surgical masks (defined as fluid-resistant or certified for surgical use) because of the relatively lower risk because all of the patients who were suspected to be infected by COVID-19 with associated symptoms listed by the WHO have been isolated in the quarantined area outside the ED [17]. All healthcare workers wear surgical masks in the visiting zone because of less likely exposure to aerosols, and N95 respirators in the critical zone, as they are suspected to be exposed to aerosols and frequently deal with high-risk procedures, such as airway fluid suctioning or endotracheal intubation for patients with respiratory failure.

### 2.2. Participants

From 1 May to 17 June 2021, 68 healthcare workers were enrolled in the study. Among them, 34 participants wore N95 respirators, and the other 34 participants wore surgical masks. The choice of facemask to wear based chiefly on the risk levels in different zones of daily duty, while personal choice was respected. Physiological parameters, including oxyhemoglobin saturation (SpO_2_), partial pressure of oxygen in arterial blood (PaO_2_), partial pressure of carbon dioxide (PCO_2_), systolic (SBP) and diastolic (DBP) blood pressures, and peripheral pulse rate (PPR) equivalent to heart rate (HR) were collected at baseline and at the 2nd, 4th, 6th, and 8th working hour for N95 respirator and surgical mask groups. All healthcare workers enrolled were on duty during the day from 8:00 to 16:00. Facemasks were donned since the beginning of the shift duty and doffed at the end of the duty. Informed consent was obtained from the participants after a thorough explanation of the details of this study. None of the recruited healthcare workers refused to participate or were excluded.

### 2.3. Demographic Characteristics of Healthcare Workers

Table 1 shows the demographic characteristics of each 34 healthcare workers in the N95 respirator and surgical mask groups. There were no significant differences in age, sex, or role of worker for both groups. The mean age of the N95 respirator and surgical mask groups was 41.2 ± 8.5 and 40.2 ± 8.4 years, respectively. Male healthcare workers and nurses accounted for higher percent-ages in this study, which were 27 (79%) and 25 (74%), and 26 (76%) and 23 (68%), respectively, for the N95 respirator and surgical mask groups, respectively. A significant difference was found between the duty of work in the two groups. In the N95 respirator group, higher percentages were found for workers at the triage station and critical zone (10 (29%) and 22 (65%)) than those in the visiting zone (2 (6%)), while in the surgical mask group, a higher percentage (21 (62%)) was found for workers in the visiting zone (Table 1).

### 2.4. Measurement of Physiological Parameters

Some clinically important physiological parameters, such as SpO_2_(%), PaO_2_ (mmHg), and PCO_2_ (mmHg), were measured using Tensor Tip MTX (Cnoga Medical Limited, Israel), which uses a fingertip detector instead of traditional invasive arterial blood gas analysis measurement [18,19,20,21]. This device has a color image sensor that is sensitive to the spectrum range of 380–1000 nm continuously that can analyze tissue pigmentation via spatial-temporal-color technique and free of air pumping. It is convenient because of its portability and lightweight. It can also measure SBP (mmHg), DBP (mmHg), and PPR (beats per minute, equivalent to HR). Other clinical parameters, such as cardiac output (L/min), hematocrit level (%), red blood cell count (ml/µL), total oxygen (ml/dL), total carbon dioxide (mmol/L), mean arterial pressure (mmHg), peripheral pulse wave and blood pressure variation total 14 bio-parameters, were recorded [22,23,24]. It met the accuracy requirements of ISO 81060-2 in 2018.

### 2.5. Symptoms Evaluation after Wearing Facemasks

We designed a questionnaire that included the most common symptoms after wearing facemasks, such as shortness of breath, anxiety, headache, dizziness, difficulty talking, fatigue, and depression, which were based on the findings of previous literature [25] and Zung Self-Rating Anxiety Scale for reference.

### 2.6. Statistical Analysis

A *t*-test was used to examine any significant differences in age between the groups wearing surgical masks and N95 respirators. A chi-squared test was used to examine any significant differences in sex, role in healthcare, duty of work, and subjective symptoms between the groups wearing N95 respirators and surgical masks.

A repeated-measures generalized linear model analysis, which applied the least-squares (LS) approach, was used to evaluate hourly changes in physiological parameters. This model enabled each physiological parameter measurement from each participant as a separate observation and was adjusted for within-participants correlations. Additionally, the duty of work was adjusted, and the interaction between the time and type of facemask was shown in this generalized linear model. Normality was tested by the Shapiro–Wilks method for analyzing the difference of dependent variable at baseline with values at 2nd, 4th, 6th and 8th hours.

All data analyses were performed using SPSS (version 19.0; IBM Corp., Armonk, NY, USA) and SAS (version 9.4; SAS Inc., Cary, NC, USA), and statistical significance was set at a two-tailed *p*-value of <0.05.

### 2.7. Ethics

This study was approved by the institutional review board of Kaohsiung Medical University Hospital (KMUHIRB-E(I)-20210059) and conducted in accordance with the Declaration of Helsinki. Data were collected anonymously and were identified. Information about the participants obtained from this study was in accordance with the principles of privacy and confidentiality.

## 3. Results

### 3.1. Physiological Data of Healthcare Workers after Wearing N95 Respirators

As in the N95 respirator, there were no significant differences in most physiological parameters at 2nd, 4th, 6th and 8th hour after wearing the respirator compared to the baseline. Some significant differences were found in SpO_2_ and PaO_2_ at 4th h (from 96.59% to 96.97% and from 85.26% to 88.03% mmHg), in SBP at 8th hour (from 133.68 to 130.53 mmHg), and in HR at 2nd, 4th, and 8th h after wearing N95 respirators (from 88.91 to 84.35, 83.68, and 80.44 beats per minute (bpm)). Significant time trends of mean changes in SBP and HR from baseline to the 8th hour was noted (−0.78 mmHg and −1.39 bpm from the baseline LS-means). However, all of these were still within the normal physiological range. (Table 2).

### 3.2. Physiological Data of Healthcare Workers after Wearing Surgical Masks

Comparison between the baseline and different donning intervals showed no significant differences, except for a noticeable decrease in HR at the 2nd and 4th hour (from 85.71 to 80.35 and 81.21 bpm); however, no significant differences were observed for HR at the 6th and 8th hour, and no significant time trends of mean changes in HR from baseline to the 8th hour were noted. The HR value during the 8 h period was still within the normal physiological range. (Table 2).

### 3.3. Physiological Data Comparison of Healthcare Workers between Wearing N95 Respirators and Wearing Surgical Masks

No significant difference in most physiological parameters was noted except for HR at 8th hour after wearing facemasks between the two groups (−8.53 bpm for adjusted work duty, *p* = 0.0079). In addition, significant differences in HR of time trend from baseline to 8 h (−2.01 beats per minute for adjusted duty of work, *p* = 0.0105) and the interaction of time and type of facemask (*p* = 0.0146) between the two groups were also noted. (Table 2, Figure 1). We also excluded 10 healthcare workers who wore N95 respirators at triage outside the hospital and only compared the other 24 healthcare workers who wore N95 respirators inside the hospital with surgical mask group (all inside the hospital). The same results of significant difference of HR at 8th hour after wearing facemasks between the two groups (−8.93 bpm for adjust-ed work duty, *p* = 0.0142) and HR of time trend from baseline to 8 h (−2.16 beats per minute for adjusted duty of work, *p* = 0.0186) were still noted.

### 3.4. Symptoms of Healthcare Workers after Wearing N95 Respirators or Surgical Masks

Healthcare workers who wore N95 respirators had a higher incidence of symptoms than those who wore surgical masks. In the surgical mask group, only one participant mentioned shortness of breath, and the other 33 healthcare workers had no obvious symptoms during their 8 h-shift in daily duty. Significant differences in the incidence of symptoms, including shortness of breath, headache, dizziness, difficulty talking, and fatigue were observed between the two groups. Such feelings of discomfort were bearable and spontaneously resolved. This may be due to physiological and psychological adaptation and compliance. All healthcare workers followed the working rules set by KMUH. None of them did not use their N95 respirators or their surgical masks due to these discomfort symptoms before completing their 8 h-shift in daily duty. (Table 3).

## 4. Discussion

### 4.1. Main Finding of This Study

In our study, the only significant change in physiological parameters in the surgical mask group was a decrease in HR at the 2nd and 4th hour after wearing it in comparison with those at the baseline status (from 85.71 to 80.35 and 81.21 beats per minute). In contrast, more significant differences in physiological parameters were found in the N95 respirator group. These differences were SpO_2_ and PaO_2_ at 4th h (from 96.59 to 96.97% and from 85.26 to 88.03 mmHg), SBP at 8th h (from 133.68 to 130.53 mmHg), and HR at 2nd, 4th, and 8th h after donning (from 88.91 to 84.35, 83.68 and 80.44 beats per minute) in comparison with those at the baseline status. Significant differences in time trend of SBP and HR from baseline to 8 h were also noted (−0.78 mmHg and −1.39 beats per minute for change from baseline). Additionally, a comparison was made between the two groups. Significant differences in HR at 8th h and time trend of HR in which the N95 respirator group had lower HR than the surgical mask group did at 8 h (−8.53 beats per minute) and at the time trend (−2.01 beats per minute). The interaction between time and type of facemask also showed a significant difference in HR (*p* = 0.0146, Table 2 and Figure 1). We also excluded 10 healthcare workers who wore N95 respirators at triage outside the hospital and only compared the other 24 healthcare workers who wore N95 respirators inside the hospital with surgical mask group (all inside the hospital). The same results of significant difference of HR at 8th h after wearing facemasks between the two groups (−8.93 bpm for adjust-ed work duty, *p* = 0.0142) and HR of time trend from base-line to 8 h (−2.16 beats per minute for adjusted duty of work, *p* = 0.0186) were still noted. These findings proved that healthcare workers who worked whether inside or outside the hospital would not confound the results of the study. This indicated that the type of facemask and wearing time affected HR. However, the changes in SpO_2_, PaO_2_, SPB, and HR were still in the normal physiological range and had no obvious harmful effects on physiological health. The increasing changes in SpO_2_ and PaO_2_ for the N95 respirator group, decreasing change in SPB for the N95 respirator group, decreasing change in HR for both groups, and lower HR in the N95 respirator group than in the surgical mask group might be due to the adaptation of facemask wearing. This adaptation was reflected in physiological conditions. Over time, the physiological parameters were inclined to a better status. Therefore, there was no evidence to prove that it was physiologically harmful to healthcare workers who wore either N95 respirators or surgical masks for an 8 h shift in daily duty.

The N95 respirator group had a significantly higher incidence of symptoms, including shortness of breath, headache, dizziness, difficulty talking, and fatigue than the surgical mask group. This discomfort may spontaneously resolve and can be proved by the better physiological conditions over time. We suspected that these symptoms mainly originated from psychological factors, since most physiological parameters were not significantly different from the baseline conditions in both groups.

Overall, the N95 respirator group did not show an obvious increase in physiological burden for an 8 h donning time in comparison with the surgical mask group. Moreover, only short-term psychological influence was noted, and the symptoms gradually disappeared. This finding might be a good reason to persuade healthcare workers to wear N95 respirators for better protection.

### 4.2. Facemasks for Healthcare Workers

Along with a noticeable increase in the number of confirmed COVID-19 cases since 15 May 2021, in Taiwan, our CDC enforced a regulation that all people should wear facemasks all the time when they leave their residences [26]. However, there are no guidelines on the type of facemasks that should be worn. A similar situation was found in KMUH that healthcare workers should wear medical-graded facemasks during working hours, but they are allowed to decide on wearing either N95 respirators or surgical masks according to their judgement and preference. It is strongly recommended that health care workers wear N95 respirators during their work in high-risk zones, including the triage outside the hospital and critical zone in which healthcare workers might be exposed to patients with unclear travel, occupational, contact, and cluster history, and those requiring aerosol-generating procedures such as suctioning of patients’ secretions from airway or endotracheal intubation for patients with respiratory failure. For healthcare workers with relatively lower risk work, such as on daily duty at triage inside the hospital and visiting zone, surgical masks were sufficient for providing protection to them. It was worldwide consensus that the N95 respirator provided better protection against almost all kinds of acute respiratory infections than surgical masks [7,8,9,10]. However, the N95 respirator was unavailable for each healthcare worker because of a global shortage due to increasing demands during the COVID-19 pandemic [27]. 

### 4.3. Comparison with Other Literature Studies

Physiologically, a study reported that long-term wearing of the N95 respirator would cause hypoxia and hypercapnia, but the other types of facemasks would not [28]. Shein et al. recruited 50 adult volunteers who wore either cloth or surgical masks and found no significant difference in transcutaneous carbon dioxide (CO_2_) (tension, and SpO_2_ between baseline measurements before donning masks, donning masks at rest, and donning masks after walking briskly for 10 min. During this trial, no episodes of hypercapnia or hypoxia were noted [29]. This finding is similar to that of our study. However, this finding could only prove that it was not harmful to healthcare workers for short-term wearing of the facemask but still lacks evidence while donning it for a longer period of time, such as 8 h shift work [29]. Rebmann et al. found that the CO_2_ level increased from baseline for healthcare workers wearing N95 respirators for two 12 h shifts, but this increase was not significant. No dynamic changes in HR, SpO_2_, or respiratory rate were found [14]. Roberge et al. conducted a one-hour treadmill walking session for 10 healthcare workers who wore N95 respirators and showed no significant increase in physiological burden in terms of HR, respiratory rate, and SpO_2_. These findings are also compatible with the results of our study. However, oxygen and CO_2_ levels from the dead space of the N95 respirator were significant below and above, respectively. The possibility of hypoxia and hypercapnia after long-term wearing of the N95 respirator remains a controversial issue [30]. Sharma et al. recruited 154 healthcare workers and found that donning of N95 respirators for 5 min would cause significant changes in cerebral hemodynamics, such as increasing mean flow velocity, decreasing pulsatile index, and increasing end-tidal CO_2_ (EtCO_2_). These dynamic changes might induce migraine and de novo headache in 25% and 80% of individuals, respectively, due to vasodilatation of the brain [31]. Headache symptoms were also observed in a few healthcare workers in our study. These effects could be improved by the use of additional powered air purified respirators (PAPR), but they are not always available in realistic conditions owing to the global shortage. Sharma et al. recruited 11 healthy volunteers and the results showed that wearing N95 respirators or valve-respirators might result in a significant increase in CO2 concentration, which exceeded the value of 0.5% by long-term (8 h) threshold limit value-time weighted average (TLV-TWA) from NIOSH but was not over short-term (15 min) limits of 3%. This phenomenon can also be mitigated by the use of PAPR. All these studies pointed out that daily use of the facemasks including the N95 respirators for 8 h should not be a concern for healthcare workers, but further studies are needed to investigate the impact on elevated CO_2_ levels with long-term wearing of facemasks [32]. This concern was answered by our study that healthcare workers wearing either N95 respirators or surgical masks during their 8 h shift in daily duty did not show hypercapnia or hypoxia. In conclusion, healthcare workers should not worry about the physiological burden of wearing facemasks for an 8 h daily working shift.

Although studies designed by different scholars tend to conclude that there are no harmful physiological effects among healthcare workers who wore either N95 respirators or surgical masks for daily duty. Other studies have reported different results. Ozdemir et al. found that donning of the PPE inclusive of N95 respirators by healthcare workers for 30 min would significantly increase the fraction of inspired CO_2_ (FiCO_2_) and EtCO_2_ without obvious symptoms. Other physiological parameters such as HR, SpO_2_, and respiratory rate were unchanged [33]. However, latent damage after long-term wearing of PPE is still a concern. Lim et al. reported that healthcare workers who wore N95 respirators for more than 4 h might cause headache in 37.3% of the participants, and 32.9% of them needed pain killers for mitigation [12]. This might have a negative effect on job performance. Fletcher et al. also reported a case of hypercapnia in an intensivist after wearing an N95 respirator. Dyspnea and tachycardia occurred while he underwent tracheostomy, which would endanger the safety of the patient due to poor performance caused by deteriorating physiological conditions [34]. These symptoms might occur more commonly among healthcare workers with a high body mass index (BMI) after wearing a facemask for two 12 h shifts [14]. In addition, some scholars considered that long-term wearing of facemasks might potentially devastate healthy conditions due to the possibility of hypoxia or hypercapnia. These might affect the balance of physiological and psychological conditions and deteriorate existing chronic diseases such as ischemic heart disease, chronic obstructive pulmonary disease, or cancer [35,36]. A study recruited 39 patients with end-stage renal disease who wore N95 respirators during hemodialysis. PaO_2_ markedly decreased (from 101 to 92 mmHg) and respiratory rate increased significantly (from 17 to 19 breaths per minute). Chest tightness and respiratory distress occurred frequently in these patients [28]. 

In addition to physiological conditions, the psychological-related effects of wearing N95 respirators or surgical masks have also been described in several studies. Chronic stress after long-term wearing along with possible hypercapnic or hypoxic conditions might affect the balance of the body and cause symptoms such as headache, dizziness, mood disturbance, and cognitive performance decrease [37,38]. These might increase anxiety and depression that deteriorated social connection and relationship, increase the risk of mortality to 40−45% according to the meta-analytic review by Holt-Lunstad J. et al. [39]. The healthcare workers in our study who wore N95 respirators had similar symptoms but no obvious hypercapnic or hypoxic condition was found. Dr. Rituparna Das shared another case report of a 52-year-old anesthesiologist with high BMI who wore an N95 respirator with a surgical mask overlay for anesthetizing the patient under surgery and experienced sudden onset of dizziness, headache, and restlessness with tremor at time close to the end of the surgery. He decided to inhale oxygen at 5 L/min by himself due to the lack of another anesthetist for alternating and symptoms subsided within 2 to 3 min. After finishing his duty, he underwent chest radiography, 12-lead electrocardiography, cardiac ultrasound, and series blood examination, and no particular abnormality was reported. These symptoms might be caused by psychological maladaptation and chronic stress after long-term wearing of N95 respirators [40]. Rebmann et al. reported that 10 nurses in a medical intensive care unit who wore N95 respirators alone or with surgical mask overlay had headache, lightheadedness, perceived shortness of air, and difficulty communicating increased over time, and 90 percent of them tolerated two 12 h shifts. Approximately 22% of them removed N95 respirators due to discomfort. However, the adaptation of N95 increased with time, and compliance was fairly high [14]. This finding was similar to that of our study, but the psychological effect depended on the subjective description of healthcare workers, and no objective measurements were taken in this study.

### 4.4. Strength of This Study

The strength of our study is that we traced the physiological parameters of healthcare workers for an 8 h shift in daily duty. This enables us to understand the dynamic changes in the physiological parameters of these healthcare workers at baseline and 2nd, 4th, 6th and 8th h after the donning of facemasks. Furthermore, this study is a pioneering study of continual measurement of normal arterial blood gases via a noninvasive method in healthcare workers who wore facemasks during their routine working shifts in the ED. Finally, significant differences between symptoms of healthcare workers after wearing N95 respirators and surgical masks were recorded via questionnaires simultaneously while physiological measurements were taken. This enabled us to evaluate the effects of the interaction between physiological and psychological conditions.

### 4.5. Limitations of This Study

Our study had some limitations. First, it was impossible for healthcare workers to wear N95 respirators or surgical masks for continuous 8 h without breaks. Generally speaking, most healthcare workers need to drink water or beverages for 2 to 3 min once an hour. In addition, during the fourth to sixth hours of daily duty, they would take turns to eat lunch for 20 to 30 min. These might have influenced the physiological parameters, but the influence was limited because of the short period of time. Second, it was difficult to ask all the healthcare workers to take rest calmly every time while their physiological parameters were taken because these healthcare workers were eager to finish the measurements quickly, and then to continue their work. Consequently, this might have influenced the physiological parameter measurements under different circumstances. Finally, only 68 healthcare workers were enrolled in our study, all of whom were on duty during the day from 8:00 to 16:00 in the same hospital. More comprehensive studies that include a larger sample size of healthcare workers, workers from different hospitals, and workers at different work shifts should be conducted in the future to thoroughly assess the dynamic changes in the physiological parameters of healthcare workers who wear facemasks for a long period of time.

## 5. Conclusions

In conclusion, healthcare workers who wore either N95 respirators or surgical masks for an 8 h shift in the emergency department showed no obvious harmful effects on their physiological health according to our study. The N95 respirator group did not increase the risk of physiological burdens, but had higher percentages of incidence for several symptoms, such as shortness of breath, headache, dizziness, talking difficulty, and fatigue in comparison with the surgical mask group. These symptoms might spontaneously resolve when healthcare workers are adaptive to wearing facemasks. The findings of this study reassure healthcare workers no significant change in their physiological and psychological health while wearing both facemasks worked on an 8 h daily duty in this pandemic era of COVID-19.

## Figures and Tables

**Figure 1 ijerph-18-13308-f001:**
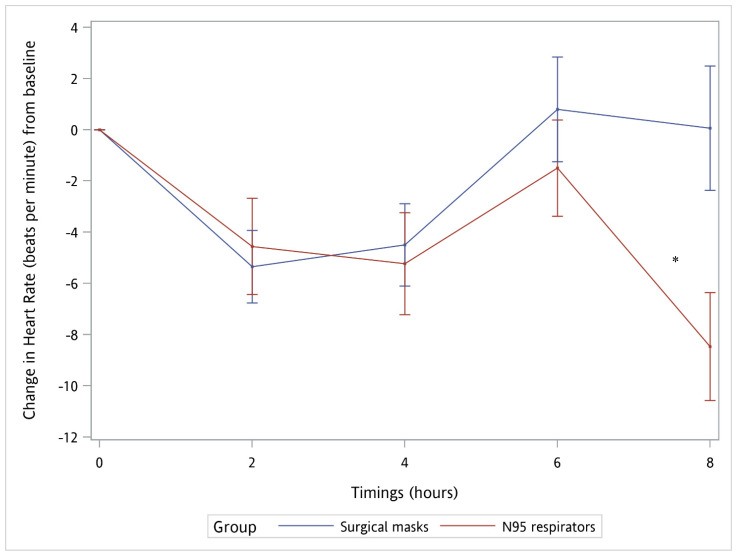
Comparison of time course of mean change in heart rate from baseline to 8 h between the N95 respirator and surgical mask groups. * The difference in heart rate trends over time of the N95 respirator group compared to that of the surgical mask group was −2.07 ± 0.79 (95% confidence interval, −3.56 to −0.47; *p* = 0.0105). The *p*-value for the interaction time and type of facemask between the two groups was 0.0146. Error bars represent the standard error.

**Table 1 ijerph-18-13308-t001:** Demographic characteristics of healthcare workers in N95 respirator and surgical mask groups.

Subject	N95 Respirator (n = 34) Mean ± SD or n (%)	Surgical Mask (n = 34) Mean ± SD or n (%)	*p*-Value
**Age (years)**	41.2 ± 8.5	40.2 ± 8.4	0.595
**Sex**			0.567
Male	27 (79%)	25 (74%)	
Female	7 (21%)	9 (26%)	
**Role in healthcare**			0.417
Nurse	26 (76%)	23 (68%)	
Doctor	8 (24%)	11 (32%)	
**Duty of work**			<0.001
Triage station	10 (29%)	5 (15%)	
Visiting zone	2 (6%)	21 (62%)	
Critical zone	22 (65%)	8 (23%)	

SD, standard deviation.

**Table 2 ijerph-18-13308-t002:** Time course of mean change in physiological parameters from baseline to after 8 h of wearing N95 respirator or surgical mask.

	N95 Respirator (n = 34)			Surgical Mask (n = 34)			N95 Respirator vs. Surgical Mask		
	LS-Means (SE)	Change from Baseline LS-Means (SE)	† *p*-Value	LS-Means (SE)	Change from Baseline LS-Means (SE)	† *p*-Value	* Adjusted Difference of LS-Means (SE; 95% CI)	† *p*-Value	‡ P-for-Interaction
§ **SpO_2_**									
Baseline	96.59 (0.13)	Reference		96.79 (0.14)	Reference		Reference		
2 h	96.82 (0.15)	0.24 (0.17)	0.1708	97.03 (0.16)	0.24 (0.19)	0.2066	0.00 (0.25; −0.50 to 0.50)	1.0000	
4 h	96.97 (0.17)	0.38 (0.17)	0.0258	96.85 (0.15)	0.06 (0.17)	0.7312	0.32 (0.24; −0.15 to 0.80)	0.1819	
6 h	96.62 (0.11)	0.03 (0.13)	0.8271	96.85 (0.14)	0.06 (0.19)	0.7515	−0.03 (0.23; −0.48 to 0.42)	0.8980	
8 h	96.85 (0.16)	0.26 (0.16)	0.1038	96.76 (0.14)	−0.03 (0.13)	0.8271	0.29 (0.21; −0.12 to 0.71)	0.1638	0.3518
Baseline to 8 h		0.03 (0.04)	0.3953		−0.02 (0.03)	0.4240	0.06 (0.05; −0.04 to 0.15)	0.2454	0.2501
¶ **PaO_2_**									
Baseline	85.26 (1.11)	Reference		87.41 (1.30)	Reference		Reference		
2 h	86.79 (1.31)	1.53 (1.52)	0.3147	88.50 (1.26)	1.09 (1.52)	0.4753	0.44 (2.15; −3.78 to 4.66)	0.8377	
4 h	88.03 (1.49)	2.76 (1.40)	0.0477	87.91 (1.39)	0.50 (1.55)	0.7478	2.26 (2.09; −1.83 to 6.36)	0.2785	
6 h	85.38 (0.94)	0.12 (1.23)	0.9238	87.26 (1.29)	−0.15 (1.74)	0.9327	0.26 (2.13; −3.92 to 4.44)	0.9012	
8 h	88.09 (1.43)	2.82 (1.54)	0.0664	86.59 (1.22)	−0.82 (1.20)	0.4941	3.65 (1.95; −0.18 to 7.48)	0.0619	0.3239
Baseline to 8 h		0.42 (0.34)	0.2160		−0.29 (0.28)	0.3082	0.71 (0.44; −0.16 to 1.58)	0.1090	0.1156
‖ **PCO_2_**									
Baseline	37.09 (0.24)	Reference		36.76 (0.26)	Reference		Reference		
2 h	36.59 (0.24)	−0.50 (0.30)	0.0995	36.29 (0.27)	−0.47 (0.31)	0.1245	−0.03 (0.43; −0.87 to 0.82)	0.9456	
4 h	35.35 (0.98)	−1.74 (0.95)	0.0673	36.65 (0.25)	−0.12 (0.28)	0.6792	−1.62 (0.99; −3.57 to 0.32)	0.1021	
6 h	37.00 (0.25)	−0.09 (0.26)	0.7353	36.74 (0.24)	−0.03 (0.36)	0.9343	−0.06 (0.44; −0.92 to 0.81)	0.8941	
8 h	36.56 (0.28)	−0.53 (0.29)	0.0640	36.71 (0.26)	−0.06 (0.24)	0.8082	−0.47 (0.37; −1.21 to 0.26)	0.2092	0.2940
Baseline to 8 h		−0.06 (0.07)	0.3373		0.03 (0.06)	0.5618	−0.10 (0.09; −0.27 to 0.07)	0.2674	0.2717
** **SBP**									
Baseline	133.68 (2.37)	Reference		129.26 (2.19)	Reference		Reference		
2 h	132.21 (2.30)	−1.47 (1.71)	0.3897	131.76 (1.81)	2.50 (2.04)	0.2193	−3.97 (2.66; −9.18 to 1.24)	0.1352	
4 h	130.94 (2.10)	−2.74 (1.73)	0.1146	127.29 (1.59)	−1.97 (1.61)	0.2199	−0.76 (2.36; −5.40 to 3.87)	0.7463	
6 h	130.71 (2.51)	−2.97 (2.07)	0.1513	127.97 (2.06)	−1.29 (1.4)	0.3546	−1.68 (2.50; −6.57 to 3.22)	0.5021	
8 h	130.53 (2.26)	−3.15 (1.41)	0.0259	129.00 (2.29)	−0.26 (1.70)	0.8764	−2.88 (2.21; −7.22 to 1.45)	0.1926	0.5603
Baseline to 8 h		−0.78 (0.34)	0.0224		−0.43 (0.39)	0.2682	−0.35 (0.52; −1.36 to 0.67)	0.5034	0.5048
†† **DBP**									
Baseline	83.82 (1.76)	Reference		80.79 (1.75)	Reference		Reference		
2 h	82.71 (1.82)	−1.12 (1.35)	0.4078	83.53 (1.38)	2.74 (1.61)	0.0889	−3.85 (2.10; −7.97 to 0.26)	0.0665	
4 h	82.06 (1.70)	−1.76 (1.36)	0.1942	79.62 (1.33)	−1.18 (1.51)	0.4358	−0.59 (2.03; −4.57 to 3.39)	0.7722	
6 h	82.24 (1.95)	−1.59 (1.46)	0.2782	79.91 (1.49)	−0.88 (1.23)	0.4737	−0.71 (1.91; −4.46 to 3.04)	0.7122	
8 h	81.59 (1.64)	−2.24 (1.29)	0.0820	80.56 (1.63)	−0.24 (1.14)	0.8364	−2.00 (1.72; −5.37 to 1.37)	0.2442	0.3968
Baseline to 8 h		−0.49 (0.29)	0.0853		−0.41 (0.26)	0.1117	−0.09 (0.39; −0.84 to 0.67)	0.8249	0.8249
‡‡ **HR**									
Baseline	88.91 (1.84)	Reference		85.71 (2.19)	Reference		Reference		
2 h	84.35 (2.04)	−4.56 (1.88)	0.0152	80.35 (2.15)	−5.35 (1.42)	0.0002	0.79 (2.35; −3.82 to 5.41)	0.7357	
4 h	83.68 (2.15)	−5.24 (1.99)	0.0086	81.21 (1.96)	−4.50 (1.61)	0.0052	−0.74 (2.56; −5.76 to 4.29)	0.7741	
6 h	87.41 (1.91)	−1.50 (1.88)	0.4258	86.50 (2.26)	0.79 (2.05)	0.6983	−2.29 (2.78; −7.75 to 3.16)	0.4097	
8 h	80.44 (2.14)	−8.47 (2.10)	<0.0001	85.76 (2.32)	0.06 (2.42)	0.9806	−8.53 (3.21; −14.82 to −2.24)	0.0079	0.0924
Baseline to 8 h		−1.39 (0.53)	0.0087		0.63 (0.58)	0.2830	−2.01 (0.79; −3.56 to −0.47)	0.0105	0.0146

LS-means, least squares means; SE, standard error; CI, confidence interval. * Adjusted difference of LS means by duty of work. † *p*-value defines the changes in physiological parameters from the baseline over time. ‡ *p*-for-interaction defines the time and type of facemask between the two groups. § Oxyhemoglobin saturation by pulse oximetry (%). ¶ Partial arterial pressure of oxygen (mmHg). ‖ Partial pressure of carbon dioxide (mmHg). ** Systolic blood pressure (mmHg). †† Diastolic blood pressure (mmHg). ‡‡ Heart rate (beats per minute).

**Table 3 ijerph-18-13308-t003:** Symptoms of healthcare workers after wearing N95 respirator or surgical mask.

Symptoms	N95 Respirator (n = 34) n (%)	Surgical Mask (n = 34) n (%)	*p*-Value
Shortness of breath	15 (44%)	1 (3%)	<0.001
Anxiety	1 (3%)	0	0.500
Headache	6 (18%)	0	0.012
Dizziness	5 (15%)	0	0.027
Difficulty talking	18 (53%)	0	<0.001
Fatigue	9 (27%)	0	0.001
Depression	3 (9%)	0	0.119

## Data Availability

The research data are not shared.

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
