# Peer review of "Comparison of Effects of N95 Respirators and Surgical Masks to Physiological and Psychological Health among Healthcare Workers: A Randomized Controlled Trial"

_ijerph, 2021, doi:10.3390/ijerph182413308_

Round 1

Reviewer 1 Report

Overview:  The authors do a thorough job of investigating the objective and subjective differences in mask types as they apply to healthcare workers.

Abstract:  Good overall abstract that provides important details of the experiment.  Line 11-12 “The incidence..” is a bit confusing and awkward.  Consider rephrasing to something like “Physiological parameters were measured via questionnaires”.  It is also confusing what was used to measure psychological parameters (if the questionnaire was used to measure psychological effects, this needs to be clarified).    Also, a small conclusion sentence needs to be included.

Introduction:  Good overview of the studies on masks and how they relate to Covid. 

Line 45-46 – is there are reason why “airway via close interpersonal contact” is bold? 

Line 66-69 – This line suggests the general need for further studies rather than stating the purpose and focus of the study in this paper.  This certainly helps to contextualize the study

Materials and Methods: The authors do an exceptional job of providing context of the participants regarding timing and PPE measures, requirements, and personal choice for masks.  They also provided enough detail as to the types of measures and equipment that were used to measure the physiological values.  The use of the t-test and chi-square test are good choices as they are more direct comparisons between the groups as opposed to other, unnecessarily complex statistical tests. 

More detail should be added for the psychological measures (section 2.3) to show the types of questions asked (was it a Likert scale or other measurements?) in addition to the reference provided. The reader needs more detail as to how these items were measured. 

Results:  The results section provides adequate information with the exception of psychological measures.

Line 177 – the word “significant” was used but no actual results are given.  Please provide measurable results or change the wording to reflect the actual findings.  Also, the word significant is used to describe a variety of symptoms so it would be best if the reader could know the individual results of each of these symptoms. 

Discussion:  Good discussion points for both the healthcare field as well as the general public with implications discussed for mask requirements in public places and what type of mask ones should wear.  The authors also did a good job of taking into consideration the limitations of the sample and different approaches that could be considered in the future. 

Conclusions: Well-written and thorough. 

Author Response

Dear Reviewer, I have revised my manuscript as your advises as possible. Detail reply is in word file below. Thanks for your comments

Your sincerely

Che-Yu Su

Reviewer 2 Report

Dear Authors:

I congratulate you on the work. I consider it a socially relevant work, it is rigorous and presents important results. However, I believe that some modifications should be made:
1. In the abstract you should include the main numerical data.
2. They should make two separate sections, one for the research design and the other for the sample.
3. The sample should be adequately described in terms of age, sex and role (nurse, medical doctor, etc.).
4. They should perform a generalizability analysis in order to ensure that the sample used produces reliable and generalizable results.
5. It should be noted whether the study was conducted in accordance with the Declaration of Helsinki (WMA 2000, Bošnjak 2001, Tyebkhan 2003), which establishes the fundamental ethical principles for research involving human subjects. 
6. Section 3.1. should be in the description of the sample.

Yours sincerely.

Author Response

(The authors gave the same response as above.)

Reviewer 3 Report

  1. Whether long-term 66 wearing of either N95 respirators or surgical masks- please define what time duration the authors meant by long term.
  2. cardiac output, hematocrit level, red blood cell count, and 14 other bio-parameters- please discuss in detail all parameters and the techniques used to measure with citations.
  3. In the N95 respirator group, higher percentages were 143 found for workers at the triage station and critical zone (29% and 65%) than those in the visiting zone (6%), while in the surgical mask group, a higher percentage (62%) was found for workers in the visiting zone (Table 1).- this is not a result, as mentioned in the introduction, this is the hospital policy to wear N95 at triage and surgical mask in visitor zone.
  4. 4. Strength of this study- There were no statistical differences found in this study then what is the translational significance; only to measure physiological and psychological symptoms- not justifiable
  5. In material and methods, the authors mentioned multiple parameters but have provided data for a limited symptom.
  6. There are various other studies evaluating physiological and psychological symptoms while using mask and comparing N95 and surgical mask with higher number of subjects. Please discuss what new this manuscript is adding to the literature.
  7. The small number of patients is a major limitation.
  8. Comparing the subjects using N95 outside hospital with surgical mask inside hospital might have confounded the results.
  9. No tables have been provided with the manuscript.

Author Response

(The authors gave the same response as above.)

Round 2

Reviewer 2 Report

Thank you for your work. The authors have responded adequately to the issues raised. It would have been interesting to perform a generalizability analysis.
Congratulations.
Best regards.

Reviewer 3 Report

Please edit the manuscript for the English language and get it checked by a native English speaker. Still there are many broken sentences and missing proper use of preposition.